# Mechanical ventilation as an independent risk factor for mortality in COVID-19-related ARDS: A secondary analysis using propensity score weighting

**David Rene Rodriguez Lima**[1,2☉]*, **Nicolás Molano-González**[2☉],
**Andrea Vargas Villanueva**[1‡], **Dario Isaias Pinilla Rojas**[1‡], **Cristhian Rubio Ramos**[1‡],
**Leonardo Andrés Gómez Cortes**[1‡], **Edith Elianna Rodríguez Aparicio**[1‡],
**Andrés Felipe Yepes Velasco**[3,4‡]

1 Critical and Intensive Care Medicine, Hospital Universitario Mayor-Méderi, Bogotá, Colombia, 2 Grupo de Investigación Clínica, Escuela de Medicina y Ciencias de la Salud, Universidad del Rosario, Bogotá, Colombia, 3 Critical and Intensive Care Medicine, Fundación Santa Fe de Bogotá, Bogotá, Colombia, 4 School of Medicine, Universidad de los Andes, Bogotá, Colombia

☉ These authors contributed equally to this work.
‡ AVV, DIPR,CRR,LAGC,EERA and AFYV also contributed equally to this work.
* drrodriguezl@hotmail.com (DRRL)

## Abstract

### Introduction

The optimal role of invasive mechanical ventilation (IMV) in COVID-19–related acute respiratory distress syndrome (C-ARDS) remains uncertain. During the pandemic, many patients with ARDS were managed without IMV, creating a unique opportunity to examine whether IMV is an independent risk factor for mortality rather than a marker of disease severity alone. This study aimed to estimate the adjusted association between IMV and in-hospital mortality in patients with C-ARDS.

### Methods

We performed a secondary analysis of a previously published prospective cohort of adults hospitalized with confirmed C-ARDS at a tertiary center located at high altitude (2,640 m, Bogotá, Colombia). Covariate balancing propensity scores (CBPS) were used to derive inverse probability of treatment weights (IPTW). Weighted logistic regression was then applied to estimate the average treatment effect (ATE) of IMV on in-hospital mortality.

As a secondary objective, respiratory mechanics during the first 5 days of IMV were described to evaluate adherence to lung-protective ventilation.

**Data availability statement:** All relevant data are within the paper and its Supporting Information file.

**Funding:** The author(s) received no specific funding for this work.

**Competing interests:** The authors have declared that no competing interests exist.

## Results

A total of 1,724 patients with complete data were included; median age was 68 years, 65.9% were male, and overall mortality was 44.8%. Of these, 897 patients (52.0%) required IMV. Mortality differed markedly between groups: 65% in ventilated patients vs. 22% in non-ventilated patients. After IPTW adjustment, IMV remained independently associated with higher mortality (ATE-adjusted OR 7.67; 95% CI 6.20–9.48; $p < 0.001$).

Respiratory mechanics were available for 838 (93.4%) ventilated patients. Median tidal volume, plateau pressure, and driving pressure were initially within protective ventilation targets; however, non-survivors showed small progressive increases in plateau and driving pressures over time.

## Conclusions

In this propensity score–weighted cohort of patients with COVID-19–related ARDS, IMV was strongly associated with in-hospital mortality after adjustment for measured confounders. Ventilatory parameters were generally within protective ranges during the early course of ventilation, although non-survivors showed less favorable longitudinal pressure trajectories. These findings support careful patient selection, optimization of non-invasive support when feasible, and strict adherence to lung-protective ventilation strategies. Residual confounding cannot be excluded.

## Introduction

Acute respiratory distress syndrome (ARDS) is characterized by diffuse inflammatory lung injury leading to hypoxemic respiratory failure and reduced respiratory system compliance [1]. Invasive mechanical ventilation (IMV) is a cornerstone of ARDS management, but it can also contribute to ventilator-induced lung injury (VILI) through mechanisms such as volutrauma, barotrauma, atelectrauma, and biotrauma [2].

The COVID-19 pandemic generated a large global population of ARDS patients and revealed marked heterogeneity in respiratory support strategies, including widespread use of non-invasive modalities [3]. Despite advances in ventilatory management, ARDS mortality remains high, typically ranging from 35% to 45%, and was even higher in COVID-19-related ARDS (C-ARDS) in several reports [4,5].

Protective ventilation strategies—including low tidal volume ventilation, appropriate positive end-expiratory pressure (PEEP), prone positioning, and neuromuscular blockade—have been associated with improved outcomes in ARDS [6–9].

The use of systemic corticosteroids became common in C-ARDS and has been associated with improved outcomes in selected populations [10,11].

During the COVID-19 pandemic, many ARDS patients were successfully managed without IMV using alternative oxygenation strategies such as high-flow nasal cannula and non-invasive ventilation. This experience contributed to updated ARDS definitions that include patients receiving non-invasive respiratory support [1]. These

observations raised questions about the optimal timing and indication for IMV and whether outcomes are primarily driven by disease severity rather than the ventilatory strategy itself [1,12,13]. Randomized trials directly comparing IMV versus non-IMV strategies in ARDS are unlikely to be feasible for ethical and practical reasons.

We aimed to evaluate whether IMV is independently associated with in-hospital mortality in COVID-19–related ARDS using propensity score weighting. As a secondary objective, we characterized early ventilatory mechanics to assess adherence to lung-protective targets.

## Methods

This study is a secondary analysis of a publicly available dataset that was previously published [4]. Ethical approval for this analysis was obtained from the Ethics Committee of Universidad del Rosario (authorization number: DVO005 1802-CV1489). Informed consent was not required, as the data were anonymized prior to analysis.

All patients who met the inclusion criteria between March 19, 2020, and July 31, 2021, were included. Data were extracted from the institutional electronic medical records. The inclusion criteria in the original study were:

1. Positive nasopharyngeal swab for *Severe Acute Respiratory Syndrome Coronavirus 2* (SARS-CoV-2).

2. Presence of bilateral diffuse alveolar infiltrates on chest X-ray or computed tomography.

3. Arterial oxygen pressure/ Fraction of inspired oxygen (P/F) ratio <300 mmHg, regardless of the oxygen delivery system; and

4. No clinical evidence of left atrial hypertension.

The definition of ARDS followed the Berlin criteria [14], incorporating the Kigali modification [15]. Thus, all patients fulfilling ARDS criteria were included, independent of whether they required invasive mechanical ventilation (IMV). Exclusion criteria included:

1. Preexisting chronic lung disease with home oxygen dependence prior to admission.

2. Lack of informed consent in the original registry; or

3. Hospital admission for a primary diagnosis other than COVID-19.

4. Incomplete data of the variables including in the propensity score.

### Data extraction

This secondary analysis was conducted using a pre-existing, de-identified dataset obtained from the institutional registry of the Mayor Méderi University Hospital in Bogotá, Colombia (2,640 meters above sea level). The data used in this study were accessed on August 15, 2025. During the study period, the hospital treated 7,345 patients with confirmed COVID-19.

The database included demographic variables (sex, age, weight, height), hematologic and biochemical parameters (complete blood count, creatinine, D-dimer, lactate dehydrogenase, C-reactive protein, electrolytes, ferritin, procalcitonin, liver enzymes, glucose, and albumin), physiological parameters (respiratory rate, heart rate, blood pressure), steroid use and disease severity indices such as the Charlson Comorbidity Index and P/F ratio.

For each patient, P/F ratio values were available at three time points: at admission, at the time of ARDS diagnosis, and the worst value recorded during hospitalization. All laboratory and physiological variables were measured within the first 24 hours of hospital admission, following the institutional protocol used in the original publication.

Only patients with complete data for all variables included in the propensity score model were eligible for this secondary analysis. The dataset was fully anonymized prior to use, and no new data collection or patient contact was performed.

 

Repeated measures were performed for the IMV data. The IMV data included: tidal volume (TV) adjusted to the predicted body weight (PBW) using the adjusted formula for males and females [males = 50 + 0.91 x (height in cm-152.4), females = 45.5 + 0.91 x (height in cm-152.4)], plateau pressure (Pplat), driving pressure (DP) that was calculated as Pplat-PEEP; and daily hours spent in the prone position. Plateau pressure (Pplat) was recorded only in patients receiving volume-controlled ventilation in whom an inspiratory pause maneuver was feasible. Mechanical power (MP) (J/min) was calculated using a simplified equation: $MP = 0.098 \times RR \times VT(L) \times (Ppeak - (Pplat - PEEP)/2)$. The IMV data were taken from Day 0, Day 1–4, considering Day 0 as the day of orotracheal intubation. The first full pulmonary mechanics during the first 6 hours after the beginning of IMV was reported for Day 0. The first respiratory mechanics were measured in the morning shift from 7 a. m. to 1 p.m. and were reported on Days 1–4. In addition, the data from the last IMV day were taken. For patients who survived discharge, this was the extubation day, and for patients who died, this was the death day. For these data, the last ventilation mechanics reported in the medical record were taken. The primary outcome evaluated was in-hospital mortality.

Sedation exposure, sedative class, and neuromuscular blocker use were not available in the dataset and could not be included in the propensity score model.

The dataset analyzed in this study is publicly available to ensure transparency and reproducibility, and the full dataset has been included as Supporting Information (S1 File).

## Statistical analysis

Qualitative variables were reported as frequencies and percentages, whereas quantitative variables were expressed as means and standard deviations or as medians and interquartile ranges, depending on the normality of their distribution (assessed with the Shapiro–Wilk test). Comparisons between survivors and non-survivors were performed using the independent samples t-test or Mann–Whitney U test for continuous variables, as appropriate, and the chi-square test for categorical variables.

To estimate the independent effect of IMV on in-hospital mortality, we applied a causal inference framework using inverse probability of treatment weighting (IPTW). This method aimed to balance the distribution of confounders between ventilated and non-ventilated patients. The covariate balancing propensity score (CBPS) method was used to estimate the propensity scores, as described by Huffman and van Gameren [16]. The selected estimand was the average treatment effect (ATE).

Subsequently, the IPTW weights were applied in a weighted logistic regression model that included all covariates used for the IPTW, incorporating interaction terms with IMV status. This model allowed estimation of the ATE through the weighted g-computation approach [17].

Longitudinal trajectories of ventilatory parameters were modeled using generalized least squares (GLS) regression with restricted maximum likelihood (REML). This approach models within-subject correlation through an explicit covariance structure rather than random effects. Fixed effects included time since initiation of IMV, survival status, and their interaction term. This framework allows unbiased estimation of population-average temporal trends while accounting for repeated measurements within patients.

Time was modeled as a continuous variable. Regression coefficients for time represent the average daily change in each ventilatory parameter among survivors (reference group), while the interaction term estimates the difference in slope between non-survivors and survivors. Model estimates are reported as β coefficients with associated p-values. These models were used to characterize differential ventilatory trajectories according to mortality outcome.

The within-subject correlation between repeated measurements was modeled using a general correlation structure, which allows each observation from the same patient at different time points to be correlated. This approach provides unbiased estimates of both fixed effects and the covariance structure.

Model specification and fitting followed the methodological framework described by Gałecki and Burzykowski [18].

All analyses were performed using R software (version 4.1; R Foundation for Statistical Computing, Vienna, Austria). Statistical significance was defined as a two-tailed *p* value <0.05.

This study was reported in accordance with the Strengthening the Reporting of Observational Studies in Epidemiology (STROBE) guidelines, and the checklist was included as Supporting Information (S2 File)

## Results

Between March 2020 and July 2021, 2,313 patients met criteria for COVID-19–related ARDS (C-ARDS). Among these, 103 lacked P/F ratio data at diagnosis, leaving 2,210 patients in the original cohort. For the present secondary analysis, only 1,724 patients with complete data for variables included in the propensity score were analyzed.

The median age of the cohort was 68 years [IQR 58–77], and 1,137 patients (65.9%) were male. The median P/F ratio at admission was 168.7 [IQR 110.0–225.3]. Among the 1,724 included patients, 773 (44.8%) died during hospitalization, 1,372 (79.5%) received systemic corticosteroids, and 897 (52.0%) required IMV.

Table 1 displays the baseline characteristic of the study population and bivariate analysis by survival status at hospital discharge.

### Propensity Score Weighted Analysis

After excluding incomplete cases, 1,724 patients with confirmed COVID-19–related ARDS were included in the propensity score analysis.

A covariate balancing propensity score (CBPS) model was constructed to adjust for baseline imbalances between patients managed with IMV and those managed without IMV. The model incorporated demographic variables (age, sex, body mass index), comorbidities (Charlson Comorbidity Index components), laboratory parameters (complete blood count, inflammatory biomarkers, renal and liver function), steroid use and severity of ARDS at admission P/F ratio).

Before weighting, patients who received IMV were older and showed higher inflammatory markers (CRP, ferritin, and LDH), more frequent steroid use, and more severe hypoxemia.

After applying inverse probability of treatment weighting (IPTW) based on the CBPS, the standardized mean differences (SMDs) for all covariates were reduced to below 0.1. Fig 1 displays the SMDs before and after weighting.

The average treatment effect (ATE) was estimated using a weighted logistic regression model in the IPTW sample. Mechanical ventilation was associated with a substantial

increase in the odds of in-hospital mortality, with an ATE-adjusted odds ratio (OR) of 7.67 (95% CI 6.20–9.48; $p < 0.001$). When expressed as a risk ratio, the ATE-adjusted relative risk (RR) was 3.18 (95% CI 2.77–3.66; $p < 0.001$), indicating that mortality was approximately three times higher among ventilated patients compared to non-ventilated patients after balancing for confounders.

The administration of IMV was evaluated by analyzing the temporal evolution of key ventilatory parameters, including tidal volume, plateau pressure, driving pressure, and duration of prone positioning. Longitudinal trends were modeled using generalized least squares (GLS) models fitted with restricted maximum likelihood (REML). Each model included time since initiation of IMV, mortality status (survivor vs. non-survivor), and their interaction term (time × mortality) as fixed effects.

Among 897 patients who required invasive mechanical ventilation, 838 (93%) had longitudinal data available for at least one ventilatory parameter and were included in the analysis. Within-subject correlation coefficients ranged from 0.2 to 0.7, indicating moderate temporal dependence across repeated measurements. Model diagnostics, including Akaike Information Criterion (AIC), Bayesian Information Criterion (BIC), and residual standard error, indicated adequate model fit. Fig 2 shows the temporal evolution of mechanical ventilation parameters according to survival status.

Table 2 displays the results of the generalized least squares models, including the estimated coefficients, their statistical significance, and the interpretation of each fixed effect.

**Table 1. Baseline characteristics of the study population and bivariate analysis according to survival status at hospital discharge (n = 1,724).**

| Domain | Variable (unit) | Survivors (n = 951) | Non-survivors (n = 773) | p-value |
|---|---|---|---|---|
| **Demographics** | Age (years) | 64 [54–74] | 71 [63–80] | <0.001 |
| | Male sex — n (%) | 642 (67.5) | 495 (64.0) | 0.144 |
| | Height (cm) | 164.4 [160–170] | 162.8 [157–170] | <0.001 |
| | Weight (kg) | 74.5 [65–81] | 72.8 [63–80] | 0.015 |
| | BMI (kg/m²) | 27.5 [24.2–29.9] | 27.4 [24.1–30.1] | 0.856 |
| | Charlson Comorbidity Index | 2 [1 –3] | 3 [2 –4] | <0.001 |
| **ARDS/Oxygenation** | ARDS severity — mild/ moderate/ severe — n (%) | 418 (44.0)/ 402 (42.3)/ 131 (13.8) | 198 (25.6)/ 359 (46.4)/ 216 (27.9) | <0.001 |
| | P/F at admission (mmHg) | 184.3 [122.6–236.1] | 155.8 [93.6–206.3] | <0.001 |
| | P/F at diagnosis (mmHg) | 180.0 [121.3–232.5] | 149.1 [93.2–200.9] | <0.001 |
| | Lowest P/F during stay (mmHg) | 118.3 [66.4–168.0] | 67.5 [48.0–72.6] | <0.001 |
| | Respiratory rate (breaths/min) | 20.5 [18 –22] | 20.8 [18 –22] | 0.111 |
| **Vital signs** | Heart rate (beats/min) | 90.8 [78–104] | 88.2 [76–101] | 0.006 |
| | Systolic blood pressure (mmHg) | 126.9 [112–137.5] | 125.8 [110–140] | 0.296 |
| | Diastolic blood pressure (mmHg) | 72.2 [64–79] | 70.7 [62–78] | 0.016 |
| **CBC** | Leukocytes (×$10^9$/L) | 10.11 [6.95–12.67] | 10.06 [6.15–12.06] | 0.895 |
| | Neutrophils (%) | 81.5 [77.0–88.0] | 81.6 [77.1–88.9] | 0.838 |
| | Lymphocytes (%) | 9.8 [6.6–14.9] | 9.3 [5.5–14.4] | 0.362 |
| | Neutrophil/Lymphocyte ratio | 8.49 [5.22–13.21] | 9.02 [5.40–16.02] | 0.001 |
| | Hemoglobin (g/dL) | 14.77 [13.6–16.0] | 14.39 [12.9–16.0] | 0.001 |
| | Platelets (×$10^9$/L) | 252 [184–303] | 222 [162–268] | <0.001 |
| **Chemistry/biomarkers** | Creatinine (mg/dL) | 1.00 [0.82–1.26] | 1.12 [0.88–1.54] | <0.001 |
| | D-dimer (ng/mL FEU) | 1,042 [626–1,853] | 1,150 [707–2,119] | 0.826 |
| | LDH (U/L) | 452 [333–527] | 522 [357–602] | <0.001 |
| | Ferritin (ng/mL)† | 1,163 [775–1,674] | 1,435 [801–2,409] | 0.008 |
| | C-reactive protein (mg/L) | 140.3 [83.4–233.8] | 145.8 [89.1–220.5] | 0.548 |
| | Procalcitonin (ng/mL)† | 0.28 [0.12–0.75] | 0.62 [0.25–1.71] | <0.001 |
| | Sodium (mmol/L) | 137.9 [135–141] | 137.0 [134–140] | 0.001 |
| | Potassium (mmol/L) | 4.52 [4.12–4.84] | 4.64 [4.15–4.99] | 0.001 |
| | Glucose (mg/dL)† | 129 [109–168] | 141 [114–192] | 0.014 |
| | Albumin (g/dL)† | 2.80 [2.50–3.10] | 2.56 [2.35–2.80] | 0.013 |
| | AST (U/L)† | 44 [32–65] | 48 [33–74] | 0.034 |
| | ALT (U/L)† | 39 [25–62] | 34 [21–56] | 0.527 |
| | Total bilirubin (mg/dL)† | 0.59 [0.44–0.85] | 0.56 [0.40–0.80] | 0.147 |
| | Bicarbonate, $HCO_3^-$ (mmol/L) | 20.2 [18.1–22.2] | 19.7 [17.4–22.0] | 0.008 |
| | $PCO_2$ (mmHg) | 30.2 [26.6–33.0] | 31.0 [26.4–33.8] | 0.034 |
| **Comorbidities** | Congestive heart failure — n (%) | 35 (3.7) | 38 (4.9) | 0.230 |
| | Myocardial infarction — n (%) | 18 (1.9) | 19 (2.5) | 0.523 |
| | Cerebrovascular disease — n (%) | 20 (2.1) | 20 (2.6) | 0.615 |
| | Chronic pulmonary disease — n (%) | 52 (5.5) | 49 (6.3) | 0.507 |
| | Diabetes (any) — n (%) | 13 (1.4) | 13 (1.7) | 0.738 |
| | Renal disease (mod–sev) — n (%) | 38 (4.0) | 26 (3.4) | 0.524 |
| **Management & outcomes** | Steroid use — n (%) | 716 (75.2) | 656 (84.8) | <0.001 |
| | Mechanical ventilation — n (%) | 311 (32.7) | 586 (75.8) | <0.001 |

*(Continued)*

**Table 1.** (Continued)

| Domain | Variable (unit) | Survivors (n = 951) | Non-survivors (n = 773) | p-value |
|---|---|---|---|---|
| | In-hospital mortality — n (%) | — | 773 (44.8) | — |

Continuous variables are expressed as median [IQR] and categorical variables as n (%). Comparisons between survivors and non-survivors were performed using the Mann–Whitney U test for continuous variables and the χ² or Fisher's exact test for categorical variables, as appropriate.

† Variables with missing data; statistics are based on available cases.

P/F = Arterial oxygen pressure/ Fraction of inspired oxygen ratio; ARDS = acute respiratory distress syndrome; BMI = body mass index; LDH = lactate dehydrogenase; AST = aspartate aminotransferase; ALT = alanine aminotransferase; CRP = C-reactive protein; IQR = interquartile range.

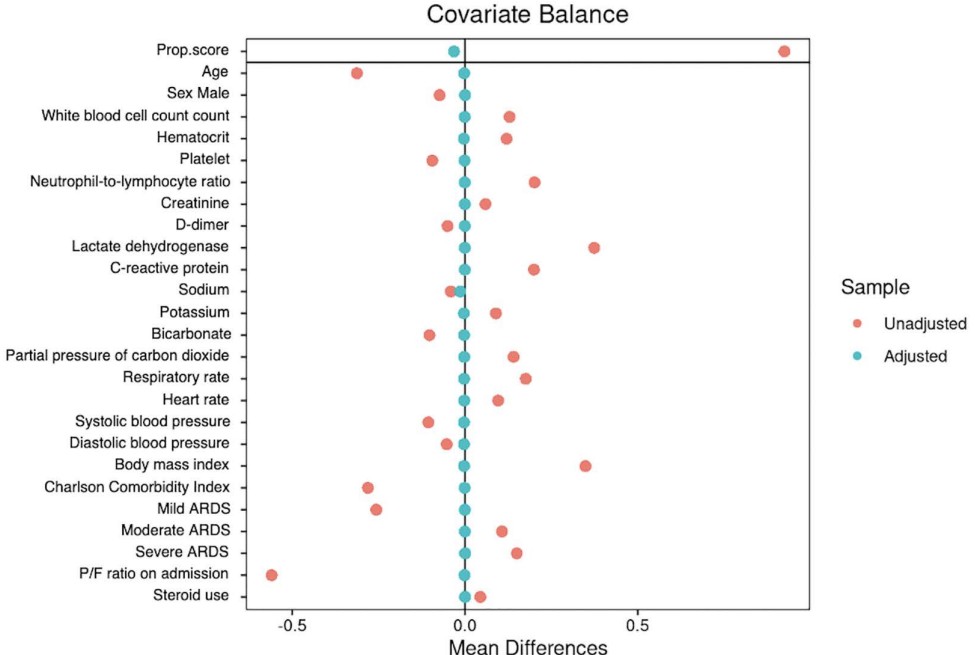

**Fig 1. Covariate balance before and after weighting.** Standardized mean differences (SMDs) for all baseline covariates are shown before and after application of inverse probability of treatment weighting (IPTW) using covariate balancing propensity scores. After weighting, all covariates achieved absolute SMD < 0.1, indicating satisfactory balance between groups.

Table 2. Fixed-effects coefficients from generalized least squares (GLS) longitudinal models evaluating temporal trajectories of ventilatory variables during the first days of mechanical ventilation. Time represents days since initiation of mechanical ventilation. Mortality compares non-survivors versus survivors (reference group). The interaction term (Time × Mortality) indicates whether temporal slopes differ between groups. TV = Tidal Volume, PBW = predicted body weight. β coefficients represent average daily change for time effects and baseline group differences for mortality effects.

Across follow-up, delivered tidal volume and PBW-adjusted tidal volume showed modest upward trends, with significantly attenuated slopes among non-survivors. Plateau and driving pressures remained stable in survivors but increased over time in non-survivors. Median values remained within lung-protective targets during the first five days of ventilation. Prone positioning duration showed greater variability and increased among non-survivors.

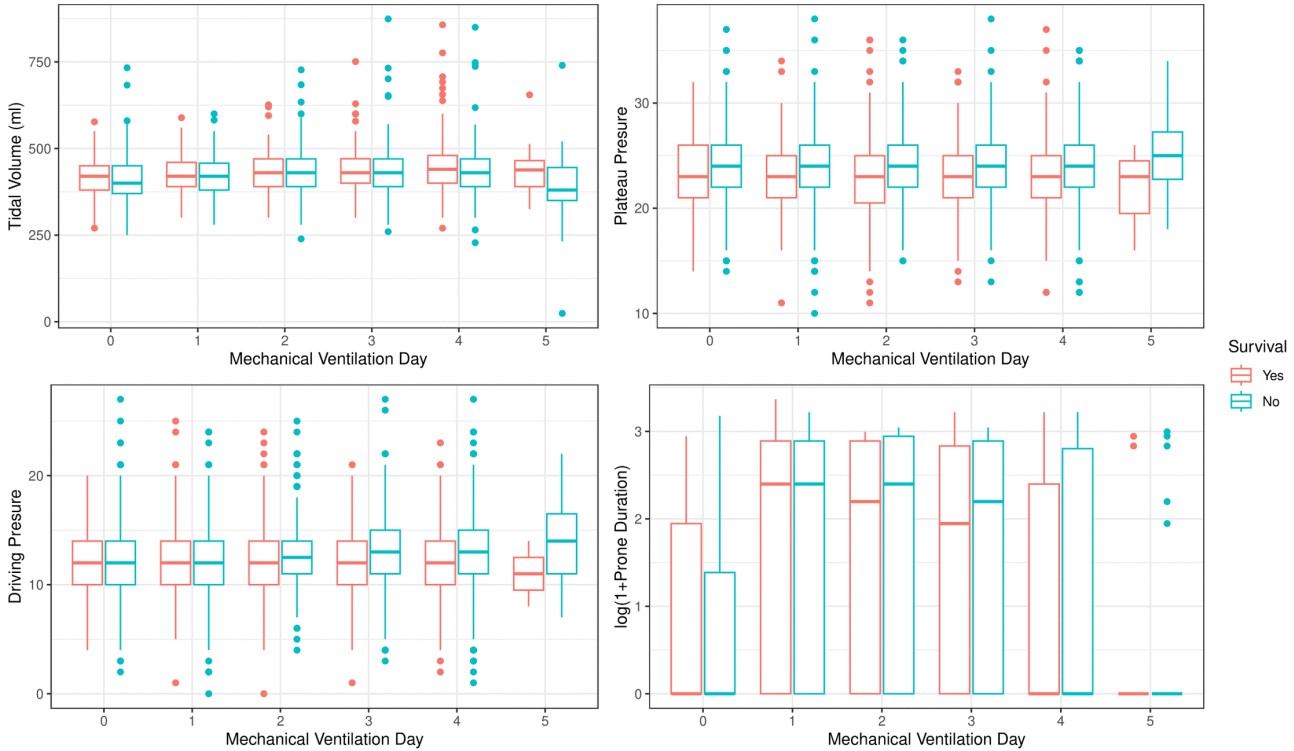

**Fig 2. Temporal evolution of mechanical ventilation parameters according to survival status.** Boxplots illustrate the longitudinal trajectories of tidal volume (upper left), plateau pressure (upper right), driving pressure (lower left), and duration of prone positioning (lower right) during invasive mechanical ventilation, stratified by survival status (alive vs. deceased). Each box represents the median and interquartile range at sequential time points, with outliers shown as individual dots.

**Table 2. Generalized least squares (GLS) longitudinal models of ventilatory parameters and mortality interaction (n = 838).**

| Outcome | Day-1 Median (Survivors) | Effect | β Estimate | p-value | Short Interpretation |
|---|---|---|---|---|---|
| Tidal Volume (mL) | 415 | Time | +6.87 | <0.001 | Daily increase in survivors |
| | | Dead vs Alive | −3.42 | 0.396 | No baseline difference |
| | | Time × Dead | −2.20 | 0.022 | Slower increase in non-survivors |
| TV adjusted to PBW (mL/kg) | 7.18 | Time | +0.127 | <0.001 | Daily increase in survivors |
| | | Dead vs Alive | +0.027 | 0.717 | No baseline difference |
| | | Time × Dead | −0.047 | 0.006 | Attenuated rise in non-survivors |
| Driving Pressure | 12.1 | Time | −0.13 | 0.070 | Stable over time |
| | | Dead vs Alive | +0.31 | 0.208 | No baseline difference |
| | | Time × Dead | +0.25 | 0.005 | Progressive increase in non-survivors |
| Mechanical Power | 12.5 | Time | +0.59 | <0.001 | Increases over time |
| | | Dead vs Alive | +0.73 | 0.041 | Higher baseline in non-survivors |
| | | Time × Dead | −0.06 | 0.620 | No trajectory difference |
| Prone Duration (log (1 + hours)) | 3.2 h (≈3 h raw) | Time | −0.018 | 0.036 | Slight overall reduction with time in survivors |
| | | Dead vs Alive | −0.093 | 0.49 | No baseline difference |
| | | Time × Dead | +0.076 | 0.013 | Progressive increase in non-survivors |

## Discussion

In this propensity score–weighted analysis, IMV was strongly associated with mortality after adjustment for measured confounders. However, as with all observational studies, residual confounding and treatment-selection bias cannot be excluded. Our findings should therefore be interpreted as association within a causal-inference framework rather than proof of direct harm from IMV itself.

The overall mortality rate for this cohort was 44.8% during hospitalization, consistent with global estimates of mortality in ventilated ARDS patients [5], which ranges from 35–50% even in C-ARDS cohorts [19]. Meta-analysis by Lim et al. [20] and Elasyed et al. [21]and Latin-American cohorts [22,23] documented similar mortality rates between 35–40%, paralleling the one observed in our study. These mortality rates indicate that despite the implementation of medical treatment and international adherence to clinical practice guidelines, patient outcomes remain unfavorable. On the other hand, consistent with global data advanced age, male sex, higher inflammatory biomarkers, thrombocytopenia, renal dysfunction and higher Charlson comorbidity scores emerged as independent mortality predictors. These factors align with large national [22,24,25] and international cohorts [26,27] amplifying IMV impact beyond the risk of either individual factor.

The first description of ARDS was published in 1967 [28], and it took almost thirty years to prove in clinical studies that IMV improperly delivered could exacerbate injury and sustain mortality, as the results in ARDS Net trial, establishing the goals of VT ≤ 6–8 mL/kg of PBW and Pplat of ≤ 30 cmH$_2$O [29–31]. Divergent longitudinal trajectories in driving and plateau pressures among non-survivors suggest worsening respiratory mechanics over time. This pattern is consistent with established VILI and P-SILI frameworks, where excessive transpulmonary stress and patient-ventilator interaction may amplify lung injury despite protective targets [32]. The longitudinal model for prone positioning duration should be interpreted with caution because this variable showed a discrete and zero-inflated distribution, which only partially satisfies generalized least squares assumptions. Accordingly, these estimates should be considered descriptive rather than strictly inferential. Variability in prone positioning exposure likely reflects real-world implementation challenges of evidence-based proning protocols, as originally demonstrated in the PROSEVA trial [33], and may relate to operational and clinical heterogeneity during the pandemic period [34,35]. However, it remains unclear whether stricter adherence to prone positioning protocols would have mitigated the suspected VILI in this cohort, given that the mean P/F ratio at the time of ARDS diagnosis was below 150 mmHg, a threshold at which proning has shown the greatest physiological and clinical benefit. In this context, delayed initiation, insufficient cumulative duration, or interruptions of prone positioning may have limited its protective effects, particularly in patients with advanced hypoxemia and high ventilatory demand.

The biological plausibility, that even with IMV protective settings susceptible patients experience progressive deterioration, rests in the well-established framework of VILI. Excessive local mechanical stress and strain disrupt alveolar epithelial and endothelial integrity, creating a high-permeability environment that amplifies cytokine release, and spreading systemic inflammation [36,37]. Furthermore, P-SILI is exacerbated by asynchronous patient-ventilator interaction due to large transpulmonary pressure swings that increase alveolar strain, worsening lung compliance and gas exchange [38]. Consistent with these findings are the higher inflammatory biomarkers as neutrophil/lymphocyte ratio, LDH, ferritin, procalcitonin, AST, and higher median DP and PaCO$_2$ values in non-survivors of this cohort.

Steroid use was more frequent among non-survivors in the crude analysis; however, this imbalance disappeared after propensity score weighting, indicating adequate covariate balance across exposure groups. This finding suggests that the unadjusted difference was driven by confounding by indication, since corticosteroids are preferentially administered to patients with greater inflammatory burden and clinical severity. Because steroid exposure was included in the propensity score specification and achieved post-weighting balance, it is unlikely to have materially biased the adjusted effect estimates. Nonetheless, residual confounding related to treatment timing, dosing, or duration cannot be fully excluded in an observational design and should be considered when interpreting treatment effects [39,40].

Our findings highlight that IMV should be approached as a precision therapy and considered a modifiable exposure, with careful attention to minimizing driving pressure, ensuring adequate gas exchange, optimizing prone positioning, and recognizing P-SILI early to prevent further harm. Therefore, non-invasive strategies such as high-flow nasal cannula and CPAP may be safely extended in selected patients and should be prioritized whenever the patient's clinical condition allows.

The major strength of this study includes a large sample size, comprehensive data validation and advanced statistical modeling. The homogeneity of C-ARDS as etiology eliminates the variability inherent to mixed ARDS cohorts, and the integration of propensity-score analysis allowed us to understand the causal pathways linking ventilation to mortality.

This study has several limitations. First, its single-center design may limit generalizability to other healthcare settings. Second, the retrospective cohort design is subject to residual bias despite propensity score weighting and multivariable adjustment. Third, residual confounding from unmeasured variables cannot be excluded, particularly factors not captured in the dataset, including timing of intubation relative to non-invasive support, sedation exposure, neuromuscular blockade use, ventilator mode transitions, and detailed ventilatory management prior to intubation.

Prone positioning duration showed a discrete and zero-inflated distribution, and related longitudinal model estimates should therefore be interpreted as descriptive rather than strictly inferential. ECMO support was not available at our institution during the study period and could not be evaluated as a rescue strategy. Finally, the high-altitude setting may limit extrapolation of these findings to sea-level populations.

In conclusion, in this large single-center cohort of patients with COVID-19–related ARDS, IMV was strongly associated with in-hospital mortality after propensity score weighting. Ventilation parameters were generally within protective ranges early after intubation, although non-survivors showed less favorable pressure trajectories. These findings support careful selection and timing of IMV, prioritization of non-invasive strategies when appropriate, and strict adherence to lung-protective ventilation. Prospective multicenter studies are needed to clarify causal pathways.

## Supporting information

**S1 File. Dataset.**
(XLSX)

**S2 File. STROBE-checklist.**
(DOCX)

## Author contributions

**Conceptualization:** David Rene Rodriguez Lima, Andrea Vargas Villanueva, Dario Isaias Pinilla Rojas, Cristhian Rubio Ramos, Leonardo Andrés Gómez Cortes, Edith Elianna Rodríguez Aparicio, Andrés Felipe Yepes Velasco.

**Data curation:** David Rene Rodriguez Lima, Nicolás Molano-González, Cristhian Rubio Ramos, Leonardo Andrés Gómez Cortes, Edith Elianna Rodríguez Aparicio, Andrés Felipe Yepes Velasco.

**Formal analysis:** David Rene Rodriguez Lima, Nicolás Molano-González.

**Investigation:** David Rene Rodriguez Lima.

**Methodology:** David Rene Rodriguez Lima, Dario Isaias Pinilla Rojas, Cristhian Rubio Ramos, Leonardo Andrés Gómez Cortes.

**Project administration:** Andrea Vargas Villanueva.

**Supervision:** David Rene Rodriguez Lima.

**Visualization:** David Rene Rodriguez Lima, Nicolás Molano-González.

**Writing – original draft:** David Rene Rodriguez Lima, Nicolás Molano-González, Andrea Vargas Villanueva, Dario Isaias Pinilla Rojas, Cristhian Rubio Ramos, Edith Elianna Rodríguez Aparicio, Andrés Felipe Yepes Velasco.

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
