## [Decision Letter · Decision Letter 0]

26 Jan 2026

Dear Dr. Rodriguez Lima,

Thank you for submitting your manuscript to PLOS ONE. After careful consideration, we feel that it has merit but does not fully meet PLOS ONE’s publication criteria as it currently stands. Therefore, we invite you to submit a revised version of the manuscript that addresses the points raised during the review process.

We look forward to receiving your revised manuscript.

Kind regards,

Jag Sunderram, M.D.

Academic Editor

PLOS One

Journal Requirements:

2. Please ensure that you refer to Figure 2 in your text as, if accepted, production will need this reference to link the reader to the figures.

3. Please update any in-text citations for your Supporting Information files at the end of your manuscript. Please see our Supporting Information guidelines for more information: http://journals.plos.org/plosone/s/supporting-information....

4. We note that there is identifying data in the Supporting Information file <PS_public_data.xlsx>. Due to the inclusion of these potentially identifying data, we have removed this file from your file inventory. Prior to sharing human research participant data, authors should consult with an ethics committee to ensure data are shared in accordance with participant consent and all applicable local laws.

-Location data

Please remove or anonymize all personal information (age), ensure that the data shared are in accordance with participant consent, and re-upload a fully anonymized data set. Please note that spreadsheet columns with personal information must be removed and not hidden as all hidden columns will appear in the published file.

Reviewers' comments:

Reviewer's Responses to Questions

**Comments to the Author**

1. Is the manuscript technically sound, and do the data support the conclusions?

Reviewer #1: Partly

2. Has the statistical analysis been performed appropriately and rigorously?

Reviewer #1: I Don't Know

3. Have the authors made all data underlying the findings in their manuscript fully available?

Reviewer #1: Yes

4. Is the manuscript presented in an intelligible fashion and written in standard English?

Reviewer #1: Yes

Reviewer #1: Overall the authors aim to present a provocative natural experiment of whether IMV independently affects outcomes from COVID ARDS. The authors do a good job describing methods and providing transparent data and data analyses. However, impact is limited by incomplete addition of variables known to affect mortality in ARDS and a lack of full description of ventilation parameter outcomes in subjects who underwent IMV.

Introduction

Too long, reads like a discussion. Suggest focusing solely on the issue of VILI and the authors main hypothesis, which appears to be that some degree of VILI is unavoidable during IMV and that use of IMV independently increases mortality.

Be careful with wording in first paragraph as decreased oxygenation does not cause reduced pulmonary compliance, per se.

Methods

While the list of variables included is long, a few key variables shown to affect mortality in COVID ARDS and ARDS as a whole are missing such steroid use (DEXA-ARDS), amount and types of sedatives used (mentioned in limitations but why couldn’t this be done?), and paralytic usage. Was awake proning prevalent in the non-intubated population?

Results

For flow I believe table 2 belongs after the propensity score weighted analyses to describe if ventilator strategies were appropriate for those who did receive IMV.

RR and mechanical power are omitted and should be included if possible.

While it is useful to state that that driving pressure was increased in non survivors over survivors, it is more useful to know if it was above 15 cmH20.

Similarity while is useful to provide the mean tidal volume, it may be moer meaningful to present this data as median ml/PBW.

The paragraph about the model for Prone Duration belongs in the discussion as part of the studies weaknesses.

Discussion

The authors speculate that mechanical ventilation worsens outcomes and admits there are unmeasured confounders, this should be tempered down the first paragraph to state that mechanical ventilation during pandemic settings with poor compliance to proning protocols may lead to deterioration. This need special attention as it was likely needed in non survivors as mean PF at diagnosis was <150.

Also it is unclear at this juncture if appropriate mechanical ventilation or inappropriate mechanical ventilation lead to mortality, for example was there a failure to achieve TV <6-8 cc PBW, driving pressure < 15, and appropriate selection for proning in on survivors?

Similarly, what was the steroid effect on outcomes? This may be especially important given that non survivors appeared to display increased inflammation.

Qualify that the single center was at high altitude with possible affects on responses to hypoxemia

Why is it not possible to measure the timing of intubation in survivors vs non-survivors? I think this is a key point. If those who died were intubated early that would add needed evidence in accord with the hypothesis, if they were intubated late, it may favor the null hypothesis.

State whether ECMO was available or not at the single institution.

.

Reviewer #1: No

---

## [Author Response · Author response to Decision Letter 1]

10 Feb 2026

Response to Reviewers

Manuscript ID: PONE-D-25-62106

Title: Mechanical Ventilation as an Independent Risk Factor for Mortality in COVID-19–Related ARDS: A Secondary Analysis Using Propensity Score Weighting

Dear Academic Editor and Reviewer,

We sincerely thank you for the careful and constructive evaluation of our manuscript. We greatly appreciate the detailed comments, which helped us substantially improve the clarity, methodological transparency, and balance of the paper. We have revised the manuscript accordingly. All changes are reflected in the revised tracked-changes version.

Below we provide a detailed, point-by-point response.

EDITORIAL REQUIREMENTS

Comment: Ensure PLOS ONE style and formatting.

Response:

We revised the manuscript to comply with PLOS ONE formatting requirements, including title page structure, figure citations in text, Supporting Information references, and file naming conventions.

Comment: Refer explicitly to Figure 2 in text.

Response:

We added explicit in-text references to Figure 2 in the Results section where ventilatory longitudinal trajectories are described.

Comment: Supporting Information citations.

Response:

All Supporting Information files are now correctly cited at the end of the manuscript following PLOS ONE guidelines.

Comment: Identifying data in dataset file.

Response:

We removed potentially identifying variables and re-generated the shared dataset with anonymized fields only, in accordance with PLOS ONE data sharing policy and ethics guidance.

REVIEWER COMMENTS – POINT BY POINT

Comment: Impact is limited by incomplete addition of variables known to affect mortality in ARDS such as steroid use, sedatives, and paralytics.

Response:

Steroid use has now been added as a covariate in the dataset description, baseline table, and propensity score model specification. Steroid exposure is reported in Table 1 and included in the CBPS/IPTW model. We added a dedicated Discussion paragraph explaining crude imbalance and post-weighting balance.

Sedation exposure and neuromuscular blocker use were not available in the dataset. This is now explicitly stated in Methods and Limitations.

Comment: Introduction too long and reads like discussion.

Response:

We substantially shortened and refocused the Introduction, removing extended historical and physiological narrative and focusing on VILI, IMV risk mechanisms, and the study hypothesis.

Comment: Physiological wording about oxygenation and compliance.

Response:

Corrected. The Introduction now states ARDS is characterized by hypoxemic respiratory failure and reduced compliance without implying causality.

Comment: Ventilator parameter table should appear after propensity score analysis.

Response:

We reordered the Results section so propensity score analysis and ATE estimates appear before longitudinal ventilatory modeling and Table 2.

Comment: Mechanical power and additional ventilator metrics should be included.

Response:

Mechanical power has been calculated using a standard simplified formula and added to Methods and Table 2. Tidal volume indexed to predicted body weight (mL/kg PBW) has also been added.

Comment: Clarify driving pressure clinical relevance.

Response:

We clarified that median values remained within lung-protective ranges early, while non-survivors showed unfavorable longitudinal trajectories.

Comment: Prone duration model caveat belongs in limitations.

Response:

Moved to the Limitations section.

Comment: Tone down causal claims.

Response:

We revised Discussion and Conclusions to emphasize association within a causal-inference framework and not proof of direct harm.

Comment: Steroid effect unclear.

Response:

We added a Discussion paragraph explaining crude imbalance, post-weighting balance, and residual confounding limits.

Comment: High-altitude setting should be clarified.

Response:

Altitude is now explicitly stated in Abstract, Methods, and Discussion.

Comment: Timing of intubation.

Response:

Not available in dataset; now explicitly listed as a limitation.

Comment: ECMO availability.

Response:

Not available in dataset; now listed as a limitation.

We thank the reviewer again for the insightful comments. We believe the manuscript is now clearer, more rigorous, and better balanced.

Sincerely,

David Rene Rodriguez Lima, MD, PhD

---

## [Decision Letter · Decision Letter 1]

23 Feb 2026

Dear Dr. Rodriguez Lima,

Thank you for submitting your manuscript to PLOS ONE. After careful consideration, we feel that it has merit but does not fully meet PLOS ONE’s publication criteria as it currently stands. Therefore, we invite you to submit a revised version of the manuscript that addresses the points raised during the review process.

We look forward to receiving your revised manuscript.

Kind regards,

Jag Sunderram, M.D.

Academic Editor

PLOS One

Journal Requirements:

Reviewers' comments:

Reviewer's Responses to Questions

**Comments to the Author**

Reviewer #1: All comments have been addressed

2. Is the manuscript technically sound, and do the data support the conclusions?

Reviewer #1: Yes

3. Has the statistical analysis been performed appropriately and rigorously?

Reviewer #1: I Don't Know

4. Have the authors made all data underlying the findings in their manuscript fully available?

Reviewer #1: Yes

5. Is the manuscript presented in an intelligible fashion and written in standard English?

Reviewer #1: Yes

Reviewer #1: Would add the following after "Variability in prone positioning exposure likely reflects real-world implementation

challenges of evidence-based proning protocols, as originally demonstrated in the PROSEVA

trial [33], and may relate to operational and clinical heterogeneity during the pandemic period

[34,35].":

It remains unclear if stricter adherence to proning protocols would have mediated the suspected injury from mechanical ventilation as the mean P/F ratio at diagnosis was <150.

.

Reviewer #1: No

---

## [Author Response · Author response to Decision Letter 2]

24 Feb 2026

Response to Reviewer #1

Manuscript ID: PONE-D-25-62106R1

Title: Mechanical Ventilation as an Independent Risk Factor for Mortality in COVID-19–Related ARDS: A Secondary Analysis Using Propensity Score Weighting

Journal: PLOS ONE

Journal requirements

All references were carefully reviewed to ensure accuracy and completeness. Reference number 28 was corrected due to a formatting error. Importantly, none of the cited references have been retracted, which was verified through PubMed and the original journal websites. No changes to the reference list involved the inclusion of retracted articles.

Reviewer #1 comment

The reviewer suggested adding the following statement after:

“Variability in prone positioning exposure likely reflects real-world implementation challenges of evidence-based proning protocols, as originally demonstrated in the PROSEVA trial [33], and may relate to operational and clinical heterogeneity during the pandemic period [34,35].”

“It remains unclear if stricter adherence to proning protocols would have mediated the suspected injury from mechanical ventilation as the mean P/F ratio at diagnosis was <150.”

Response

We agree with the reviewer’s suggestion. The requested statement has been incorporated into the Discussion section to acknowledge the uncertainty regarding whether stricter adherence to prone positioning protocols could have mitigated ventilator-associated lung injury, particularly in a cohort with severe hypoxemia at diagnosis (mean PaO₂/FiO₂ <150 mmHg).

All modifications introduced in response to this comment have been highlighted in red in the revised manuscript.

Sincerely,

David Rene Rodriguez Lima, MD, PhD

---

## [Editor Report · Decision Letter 2]

27 Feb 2026

Mechanical Ventilation as an Independent Risk Factor for Mortality in COVID-19-Related ARDS: A Secondary Analysis Using Propensity Score Weighting

PONE-D-25-62106R2

Dear Dr. Rodriguez Lima,

We’re pleased to inform you that your manuscript has been judged scientifically suitable for publication and will be formally accepted for publication once it meets all outstanding technical requirements.

Kind regards,

Jag Sunderram, M.D.

Academic Editor

PLOS One
---

## [Editor Report · Acceptance letter]

PONE-D-25-62106R2

PLOS One

Dear Dr. Rodriguez Lima,

I'm pleased to inform you that your manuscript has been deemed suitable for publication in PLOS One. Congratulations! Your manuscript is now being handed over to our production team.

Kind regards,

on behalf of

Dr. Jag Sunderram

Academic Editor

PLOS One